# EP300 as a Molecular Integrator of Fibrotic Transcriptional Programs

**DOI:** 10.3390/ijms241512302

**Published:** 2023-08-01

**Authors:** Karla Rubio, Alejandro Molina-Herrera, Andrea Pérez-González, Hury Viridiana Hernández-Galdámez, Carolina Piña-Vázquez, Tania Araujo-Ramos, Indrabahadur Singh

**Affiliations:** 1International Laboratory EPIGEN, Consejo de Ciencia y Tecnología del Estado de Puebla (CONCYTEP), Instituto de Ciencias, Ecocampus Valsequillo, Benemérita Universidad Autónoma de Puebla (BUAP), Puebla 72570, Mexico; 2Laboratoire IMoPA, Université de Lorraine, CNRS, UMR 7365, F-54000 Nancy, France; 3Departamento de Biología Celular, Centro de Investigación y de Estudios Avanzados del Instituto Politécnico Nacional (CINVESTAV-IPN), Ciudad de México 07360, Mexico; 4Emmy Noether Research Group Epigenetic Machineries and Cancer, Division of Chronic Inflammation and Cancer, German Cancer Research Center (DKFZ), 69120 Heidelberg, Germany

**Keywords:** histone acetyltransferase, fibrosis, EMT, TGFβ signaling pathway, intrinsic disorder domains, bromodomain inhibitors, EP300 degraders, epigenetics

## Abstract

Fibrosis is a condition characterized by the excessive accumulation of extracellular matrix proteins in tissues, leading to organ dysfunction and failure. Recent studies have identified EP300, a histone acetyltransferase, as a crucial regulator of the epigenetic changes that contribute to fibrosis. In fact, EP300-mediated acetylation of histones alters global chromatin structure and gene expression, promoting the development and progression of fibrosis. Here, we review the role of EP300-mediated epigenetic regulation in multi-organ fibrosis and its potential as a therapeutic target. We discuss the preclinical evidence that suggests that EP300 inhibition can attenuate fibrosis-related molecular processes, including extracellular matrix deposition, inflammation, and epithelial-to-mesenchymal transition. We also highlight the contributions of small molecule inhibitors and gene therapy approaches targeting EP300 as novel therapies against fibrosis.

## 1. Introduction

Fibrosis is a complex and debilitating condition that results from the excessive accumulation of extracellular matrix (ECM) proteins in tissues, leading to organ dysfunction and failure [1,2,3]. Despite its prevalence and significant impact on patient quality of life, there are currently no effective treatments against this disease, there are few—if any—treatment strategies available that specifically target the pathogenesis of fibrosis [1,4]. However, recent research has identified epigenetic regulation as a potential approach to target multi-organ fibrosis, particularly through targeting of histone acetyltransferases such as EP300 [5,6,7,8,9]. EP300 is a multifunctional protein that plays a critical role in the regulation of gene expression through the acetylation of histones, which alters global chromatin structure and modulates transcriptional activity. EP300 has been implicated in the pathogenesis of fibrosis, particularly through its role in promoting ECM deposition, inflammation, and epithelial-to-mesenchymal transition [6,7,8,9,10,11]. As a result, EP300-mediated epigenetic regulation has emerged as a promising target for the development of new therapies. Here, we will explore the mechanisms of EP300-mediated epigenetic regulation of fibrotic events, as well as the potential of targeting EP300 for anti-fibrotic therapies. Firstly, we will provide a brief overview of the current understanding of fibrosis and its impact on patients, highlighting the need for new and effective treatments. We will then discuss the role of EP300 in epigenetic regulation and its potential as a therapeutic target. This will include a detailed examination of the preclinical evidence that suggests EP300 inhibition can attenuate fibrosis-related processes, including ECM deposition, inflammation, and epithelial-to-mesenchymal transition. Next, we will discuss the potential of small molecule inhibitors and gene therapy approaches targeting EP300 as potential therapies for fibrosis. We will examine the preclinical evidence supporting the effectiveness of EP300 inhibitors, including their ability to reduce fibrosis-related processes in animal models of disease. We will also explore the potential challenges associated with targeting EP300, such as off-target effects and the need to optimize the potency and selectivity of EP300 inhibitors. Overall, this review aims to elucidate the potential of EP300-mediated epigenetic regulation as a therapeutic target for fibrosis. By exploring the mechanisms of EP300-mediated epigenetic regulation and its role in fibrosis, as well as the potential of targeting EP300 for the treatment of fibrosis, we hope to inspire further research and development of new and effective therapies for this devastating condition.

## 2. Pathogenesis of Fibrosis

When tissue injury occurs, a process toward its repair begins. This process relies on the recruitment of inflammatory cells to the injury site with the primary goal of wound closure, as first achieved by tissue scarring. Depending on the tissue, repairing it can almost completely restore the tissue architecture, as in the case of epithelial tissues. In a mild injury with no compromise of the basal membrane, the scarring tissue will gradually degrade over weeks and be replaced due to the action of pluripotent stem cells on the basal membrane [12,13]. Even though full architecture restoration has only been observed in human fetal skin models [14]. In a restrictive manner, in most tissues, or when injuries effectively disrupt their architecture, repair will be based on ECM components. Therefore, poorly regulated processes of inflammatory cell clearance and aberrant deposition of ECM components may lead to a dysfunctional tissue architecture with altered oxygen supply, which is a hallmark of fibrotic diseases [15]. This scenario will occur when ECM production is higher compared to the ECM degradation rate. Although pathologic repair can occur virtually in each type of tissue in the human body [16], here, we discuss renal, cardiac, lung, and liver fibrosis, which represent almost 45% of deaths in developed countries [17].

As an early event in tissue repair, inflammatory cell recruitment depends mostly on circulating monocyte progenitors, which are differentiated to an M1 profile when stimulated by interferon gamma (IFNγ) or tumor necrosis factor (TNF). When activated, immune cells highly express factors that promote fibroblast activation. Even though the specific immune cells and the mediators promoting fibroblast activation vary from one fibrotic disease to another, a common feature is the type 2 immune response, more importantly via the secretion of interleukin 13 and 14 (IL-13 and IL-4) produced by T-helper type 2 (TH2) cells, which in turn leads to M2-polarized macrophages with a subsequent release of profibrotic molecules that promote the activation and accumulation of myofibroblasts [18,19]. Myofibroblasts refer to a cellular phenotype characterized by stellate-shaped cells with high levels of intracellular proteins such as non-muscle myosin, alpha smooth-muscle actin (αSMA), and vimentin and differentiated from diverse sources (pericytes, endothelial cells, fibrocytes, epithelial cells, and resident fibroblasts). When not cleared after a scarring physiological process, they can become chronically activated with a major production of EMC components [20].

As proposed by Mehal et al. in 2011, the core pathways of fibrosis are those that are necessary to determine that one stimulus drives fibrosis and are conserved among different tissues [21]. Thus, it has been established that transforming growth factor beta (TGFβ) represents a master regulator of tissue repair responses, both in physiological and fibrotic states. In consequence, TGFβ leads to the activation of fibrotic transcriptional programs, which can be understood as sets of genes sharing regulatory mechanisms or biological functions, which in turn leads to a fibrotic response [22,23]. The evoked fibrotic signaling is so strong that the activation via TGFβRI receptor can lead to a systemic fibrotic disease with aggressive migration and invasion phenotypes through canonical TGFβ–SMAD, ERK mitogen-activated protein kinase, JNK and p38 MAPK, FAK signaling, and Rho/ROCK/NF-kappa B signaling pathways, among others [24,25,26] (Figure 1). The stiffness caused by the deposition of the ECM components is sensed by αVβ integrins, hence activating the RhoA/ROCK-YAP/TAZ axis that leads to fibrotic transcriptional programs [27], including the TGFβ target gene *SERPINE1*.

Signaling initiated by 5-hydroxytyrosine (5-HT) receptors is enhanced in fibrotic states and leads to an increased transcription of *TGFβ* as a positive loop. TGFβ signaling also leads to phosphorylation of FLI1, releasing the *COL1A2* promoter, and inducing ECM deposition. In turn, TGFβ activates kinases such as JNK, JAK2, ABL, and SRC, which phosphorylate the profibrotic protein STAT3 [28]. Developmental pathways, such as Hedgehog, are in turn activated via induction of GLI2 or inhibited by WNT repressors such as SFRP1 and DKK1 [29]. In order to counteract the fibrotic state, sGC-PKG signaling is activated and remains specific for ERK signaling blockade [30]. As for nuclear receptors, NR4A1, VDR, and PPARγ have overall inhibitory effects on TGFβ signaling. NR4A1 forms a repressor for the SP1-dependent profibrotic genes. VDRs act by competing with SMAD4 for the binding of SMAD2/3 dimers, limiting TGFβ fibrotic transcriptional programs [3].

TGFβ regulates the interaction of SMAD3 and EP300 in a temporal and phosphorylation-dependent manner [31]. Also, TGFβ regulates the expression of *EP300* via Early Growth Response 1 (EGR1) and can regulate its translocation through posttranslational modifications. For instance, some reports suggest that the phosphorylation of EP300 by AKT signaling triggers its translocation to the nucleus during liver fibrosis. Once in the nucleus, EP300 collaborates with TGFβ-activated SMAD3 on the collagen gene promoter, thus inducing collagen synthesis [31,32,33].

## 3. EP300 Functional Domains, Enzymatic Activity, and Cellular Distribution

EP300, also known as P300, is a crucial chromatin regulator of approximately 300 kDa in size, first described as an adenovirus E1A-associated protein and an adenoviral oncogenic transcription factor [33,34]. EP300 is a multifunctional protein with acetyltransferase activity for histone and non-histone targets, such as numerous transcription factors [35]. Importantly, EP300 acts as a transcriptional coactivator for several transcriptional factors, influencing multiple biological pathways.

EP300 and the CREB binding protein (CBP) are highly homologous proteins; given their functional redundancy, they are named CBP/EP300. However, in some specific circumstances, EP300 and CBP play distinct roles. For instance, it was recently shown that EP300, but not CBP, is responsible for maintaining H3K27ac at regulatory elements in mESCs. Also, distinct gene sets are transcriptionally dysregulated upon depletion of EP300 or CBP [36].

EP300 is a multidomain protein divided into a catalytic core, interaction domains, and a large set of inter-domain intrinsically disordered regions (IDRs, Figure 2). The catalytic core comprises a bromodomain, a RING-PHD domain (also known as CH2), and a histone acetyltransferase (HAT) domain [37]. The bromodomain recognizes patterns of multiple acetylation in lysine residues, and it is required for effective substrate acetylation since loss of the bromodomain impairs CBP/EP300 HAT substrate specificity and transcriptional activity. For instance, it is uniquely critical for the acetylation of histone H3 at lysine 27 at enhancers [38]. The CH2 region contains a discontinuous PHD domain interrupted by a RING domain. Instead of a canonical function, this RING domain has an inhibitory role over its HAT activity. In an inactive state, the RING domain blocks the HAT substrate-binding pocket, a displacement that is probably coupled to autoacetylation in trans and results in full catalytic activation [37].

EP300 HAT domain also contains a 60 amino acid intrinsically disordered autoinhibitory loop (AIL, residues 1520–1581, Figure 2). When hypoacetylated, it serves as a pseudo-substrate that inhibits acetyltransferase activity by competing with positively charged substrates for binding to the active site but releases the inhibition upon hyper auto-acetylation [39]. Next, EP300 (not excluding a similar mechanism for CBP) activation is initiated by enhancer recruitment of at least two copies of EP300, resulting in in-trans AIL autoacetylation and displacement of the RING domain with consequent activation of EP300 acetyltransferase activity [37].

Small, structured domains TAZ1, TAZ2, KIX, and NCBD act as scaffolds for intrinsically disordered activation domains of cellular transcription factors and other regulatory proteins, including TP53, hypoxia-inducible factor 1α (HIF-1α), NF-κB, and STAT proteins [40]. ZZ-TAZ2 domains are in the CH3 region, where the ZZ domain of CBP/EP300 recognizes histone H3 tail and therefore represents a novel epigenetic reader, promoting acetylation of primarily histone [41]. Recently, it was proposed that the TAZ2 domain plays a role in allosteric HAT regulation since binding to transcription factor activation domains (TF-AD) results in displacement of the TAZ2 domain from its auto-inhibitory position, resulting in HAT activation, thus directly coupling TF-AD binding to HAT activation [42].

EP300 is localized in the cytosol, vesicles, and nucleoplasm (Human Protein Atlas, 2023). Acetyltransferase enzymes catalyze the covalent transfer of the acetyl group from acetyl-CoA to the epsilon amine of the lysine side chain. The loss of the positive charge can change intramolecular protein conformation or affinity for binding partners [43]. EP300 and CPB acetylate sites on all four core histones, but to a differing level and site specificity: H3K18, H3K27, H3K36, and N-terminal H2B and H2A [35]. Indeed, using mouse embryonic fibroblasts (MEFs), it was found that up to one-third of the nuclear acetylome is regulated by cooperating CBP/EP300. Thus, both HATs target hundreds of acetylation sites on over 200 transcription factors, chromatin remodelers, and transcriptional co-activators.

Among them, we find more than three dozen enhancer-associated transcriptional regulators, such as the BCL9L-Wnt enhanceosome, the cohesion complex, the Mll3/4 complex, the mediator complex, the super elongation complex (SEC), and components of the general transcription machinery. CBP/EP300 acetylates nearly all members of the bHLH-PAS family and many transcriptional effectors in signaling pathways relating to development and differentiation, including Wnt-beta catenin, Hippo, Hedgehog, Notch, and TGFβ signaling pathways.

Moreover, EP300 regulates proteins in pathways involved in nutrient and energy metabolism, such as AMPK, PKA, and calcineurin signaling, among others [35]. Importantly, substrate acetylation mediated by EP300 occurs independently of linear amino acid motifs; instead, it acetylates most sites on targeted proteins. This tendency to acetylate closely spaced lysine clusters has prompted the expert researchers to propose a model where CBP/EP300 acts as an ‘‘acetyl-spray’’ at active enhancers, targeting accessible regions on proximal proteins. Such a high-density acetylation model on proximal sites could, in addition, facilitate bromodomain (BRD) protein recruitment, having an additive effect on transcription [35]. Furthermore, CBP/EP300 are also known to possess ubiquitin E3 and E4 ligase activity in a fragment spanning residues ~200–595, that cooperates with the E2 UbcH5 (E2 ubiquitin-conjugation enzyme) [44]. Notably, their ubiquitin ligase activities are absent in nuclear extracts and are exclusively cytoplasmic. In turn, cytoplasmic CBP/EP300 E3/E4 activities ubiquitinate and destabilize TP53, while nuclear CBP/EP300 activate TP53 by acetylation [45].

To date, and due to the recent progress in the dynamic field of liquid–liquid phase separation (LLPS), the EP300 structure–function relationship has been explored from a more biophysical perspective. For instance, it has been shown that the approximate 60% of the EP300 structure corresponding to IDR domains in both N- and C-terminus, as predicted by sequencing-based methods (Figure 2), direct partner interactions but also to induce the formation of membraneless transcriptional co-condensates or clusters [46,47] with other transcription factors in order to achieve high EP300 local concentrations to promote efficient transactivation and EP300 catalytic activity at specific regulatory regions such as enhancers or super-enhancers [48].

## 4. Multiorgan EP300 Dysregulation

### 4.1. Renal Fibrosis

Tubulointerstitial renal fibrosis is a progressive detrimental connective tissue deposition on the kidney parenchyma that induces renal disfunction. During renal fibrosis, renal epithelial cells are transformed into mesenchymal fibroblasts, which migrate to adjacent interstitial parenchyma [49]. It is associated with glomerulosclerosis, tubulointerstitial fibrosis, and loss of renal parenchyma (e.g., tubular atrophy, loss of capillaries, and podocytes), activation of proinflammatory pathways, as well as impairment of endogenous protective mechanisms in the kidney [50]. Several research groups have focused on the genetic and epigenetic aspects that involve the initiation and progression of renal fibrosis. In 2022, Gong et al. detected higher protein levels of EP300, HIF2α, VEGFA, αSMA, and fibronectin in human kidney tissue samples from diabetic nephropathy compared to controls, suggesting them as indicators of kidney fibrosis stage. On the other side, they identified that the silencing and the overexpression of *EP300* reduced and increased the fibrotic phenotype, respectively, through *HIF2α* expression regulation in human renal tubular epithelial cells (HK-2) stimulated with high glucose [39].

Another study demonstrated the strong association between lysyl oxidase (*LOX*) upregulation and the overall ratio of insoluble to soluble collagen in para-carcinoma tissues and renal biopsies [51]. LOX is an extracellular copper-dependent monoamine oxidase that catalyzes the cross-linking of soluble collagen and elastin into insoluble mature fibers. This result was further confirmed in murine models of renal fibrosis induced by Ischemia-reperfusion injury (IRI), unilateral urethral obstruction (UUO), and folic acid (FA) treatment. In particular, they identified via RNA sequencing analysis necessary transcription factor-mediated gene expression programs enriched in IRI, including NFKB1, JUN, STAT3, RUNX2, EP300, SPI1, CEBPB, and RFX5 [52].

In the same fashion, the transcriptomic profile of biopsy samples from 42 kidney transplant recipients at four different time points (before implantation, shortly after the restoration of blood flow in the graft, and 3 and 12 months after transplantation) was evaluated. Interestingly, profibrotic genes such as *COL1A2*, *DCN*, and *MMP2* were upregulated in a later phase, whereas genes responsive to acute kidney injury such as *HAVCR1* and *LCN2* and to innate immunity such as *TRAF6* and *TLR3* were upregulated in an early phase in human and murine samples. Remarkably, network analyses suggested EP300 to be an early-phase regulator of a cluster of 33 genes widely associated with renal function in both patient and mouse models. These findings indicated a significant role for EP300 in fibrosis initiation [53].

Additionally, epigenome-wide association studies of estimated glomerular filtration rate (eGFR) and chronic kidney disease (CKD) were performed using whole-blood DNA methylomes from the Atherosclerosis Risk in Communities (ARIC) Study with African American participants (N = 2264) and the Framingham Heart Study (FHS) with European participants (N = 2595). This research showed that hypermethylation at the eGFR-associated CpG site cg19942083 is involved in lower renal fibrosis and *PTPN6* expression, suggesting a favorable role for these components in the kidney function profile. In contrast, higher *PTPN6* expression enhanced the fibrotic phenotype. In addition, binding sites for EP300, CEBPB, and EBF1 were identified in eGFR-associated CpG sites, with the consequent regulation of endocrine, metabolic, and cell remodeling pathways.

In a more recent work, Liu et al. demonstrated that the restoration of the activity of the salt-inducible kinase SIK2 is a promising therapy for diabetic patients with interstitial fibrotic kidney disease, since the underlying molecular mechanisms involve the EP300-dependent SIK2 inactivity in renal tubules of patients and in vivo models using vancomycin-induced kidney disease in murine experiments [54]. Consistently, previous groups linked the SIK2-mediated inhibition of EP300 with a phosphorylation-based modification on Ser89, as it has been recently validated by ex vivo approaches [7]. In summary, these approaches characterized the epigenetic landscape of kidney fibrosis and validated the role of EP300 in kidney function. However, some limitations included a small sample size, cross-sectional design, replication issues, and existing comorbidities [55].

### 4.2. Cardiac Fibrosis

Myocardial fibrosis is associated with abnormal cardiac remodeling and cardiovascular diseases, including arrhythmias, atherosclerosis, hypertension, and heart failure. In cardiac fibrosis, fibroblasts proliferate and transform into myofibroblasts, with a consistent overproduction of collagen and other ECM proteins. Fibrosis increases myocardial stiffness and stress, inducing congestive heart failure and diastolic dysfunction [56]. Excess deposition of ECM between cardiomyocyte layers may result in slow conduction (electric coupling disruption) due to partial depolarization of cardiomyocytes, which is implicated in impaired myocardium contractility and arrhythmias via non-uniform impulse conduction [57,58].

Atrial fibrillation (AF) is a common cardiac aging-related arrhythmia. In order to elucidate the molecular mechanisms of age-related atrial fibrosis, an in silico analysis in young (18–49 years old) and old (≥50 years old) patients with ≥6 months of AF was performed. The authors observed that atrial tissues in older AF patients, as reproduced in aged mice and senescent human atrial fibroblasts, displayed severe AF and correlated to higher levels of EP300, acetylated TP53, SMAD3, and other phosphorylated SMADs, as well as higher levels of pro-fibrotic factors (e.g., COL1A1/3A1, MMP-2/9, and TGFβ). Curcumin treatment and genetic abrogation of EP300 rescued senescence and fibrosis effects in human atrial fibroblasts [59]. Furthermore, mature mice improved their electrophysiologic characteristics and attenuated AF susceptibility, confirming that in this model, EP300 may modulate senescence and fibrosis by protein acetylation [59]. Importantly, recent evidence suggest the participation of the monocarboxylate transporter MCT as an intermediate between EP300 and TGFβ/SMAD via SNAIL1 lactylation during cardiac fibrosis [60].

In a recent study, a decrease in EP300/CBP-associated factor (*Pcaf*), a profibrotic mediator of TGFβ also known as K acetyltransferase 2B, was reported in cardiomyocytes but upregulated in fibroblasts from mice with isoproterenol (ISP)-induced cardiac fibrosis, with a concomitant increased expression of fibrotic markers (e.g., *Acta2* and *Col1a1*) [61]. Lower PCAF protein amounts were validated in TGFβ1-treated cardiomyocytes in a time- and dose-dependent manner, while its activity was remarkably increased in treated fibroblasts. Additionally, their PCAF loss-of-function experiments using specific siRNA probes suggested that SMAD2 phosphorylation and nuclear translocation in TGFβ-treated cardiac fibroblasts are directly mediated by PCAF [61].

Furthermore, recent work in C57BL/6 male hypertensive mice treated with EP300 inhibitors (L002 and C646) showed a significant reversal of cardiac hypertrophy, fibrosis, hypertension-induced H3K9 acetylation changes, and myofibroblast differentiation in a TP53 activation-dependent mechanism [5]. Also, in 2022, Zeng et al. reported the nuclear overexpression of *NPPA-AS1*, a cardiac-enriched and conserved lncRNA located in the antisense strand of the atrial natriuretic peptide (*NPPA*) gene, in cardiomyocytes isolated from transverse aortic constriction (TAC) cardiac model in mice and in patients with heart failure too [52].

The inactivation of *NPPA-AS1* by CRISPR-Cas9 minimized TAC-induced cardiac hypertrophy and fibrosis, leading to cardiac size, weight, and function restoration. The authors proposed a mechanism that consists of GATA4 direct binding to *NPPA-AS1* as a scaffold for EP300 recruitment, promoting subsequent *GATA4* acetylation and pathological cardiac hypertrophy. It is important to mention that pharmacological inhibition of *NPPA-AS1* could be a potential cardiac-selective target in heart failure therapy because this cardiac-enriched hypertrophy-associated lncRNA is slightly expressed in other organs or tissues [62].

### 4.3. Lung Fibrosis

Pulmonary fibrosis (PF) is a chronic and progressive interstitial lung disease in which lung parenchyma is replaced by scar tissue. Frequently, it culminates in lung transplantation or death due to respiratory failure, therefore having a poor prognosis. On the other side, recurrent environmental injuries to the lung epithelium affect alveoli elasticity and promote the release of proinflammatory cytokines, proteases, and growth factors, even inducing tissue hypoxia. Different forms of PF are truly idiopathic (referred to as IPF). However, factors that contribute to PF development are not mainly genetic but a combination of autoimmune, toxicological, and environmental (by drugs, toxic agents, or ionizing radiation) [50,63].

PF is characterized by increased deposition of fibrillar collagen in concert with proteoglycans and glycosaminoglycans (e.g., versican, decorin), elevated levels of the cross-linking enzyme LOXL2, and low-molecular-weight hyaluronan, which promote the release of cytokines and stimulate chemotaxis in the ECM, specifically in fibroblastic foci. Furthermore, the accumulation of membrane-bound hyaluronan synthases in areas of tissue injury is associated with ECM production and myofibroblast differentiation. In this manner, compositional changes in ECM in lung fibrosis directly affect the architecture of the lung and promote a profibrotic environment [64,65].

Several research groups have focused on the epigenetic mechanisms that mediate the maintenance of the lung fibrotic phenotype. For instance, it has been demonstrated that HDAC7 positively regulates connective tissue growth factor (*CTGF*) and *α-SMA* expression in ET-1-stimulated normal human embryonic lung fibroblast cell lines (WI-38) [66]. Precisely speaking, ET-1 promoted the nuclear translocation of HDAC7 to establish a transcriptional complex with EP300 and AP-1, which mediated *c-JUN* acetylation by EP300 and fibrosis in the ovalbumin-induced airway fibrosis mouse model. Then, the complex is recruited to the *CTGF* promoter region, increasing *CTGF* expression [66]. Moreover, the upregulation of Discoidin domain receptor 1 (*DDR1*), a transmembrane collagen receptor, and *EP300* was observed in samples from healthy or idiopathic pulmonary fibrosis (IPF) patients by single-cell RNA sequencing [52].

Also, differentially expressed genes were enriched in autophagy, cellular response to organonitrogen compounds (including *EP300* and collagen), and metabolic (including *DDR1*) pathways. Genetic abrogation as well as pharmacologic inhibition of EP300 showed that EP300 was induced by TGFβ1-mediated FN1 and DDR1 protein levels in human lung fibroblast MRC-5 cells in a reversible manner. Additionally, the combination of EP300 inhibitor SGC-CBP300 and DDR1 inhibitor CQ-061 attenuated fibrotic (e.g., FN1 and COL1A1) and inflammatory (e.g., IL-4 and IFN-γ) effects in bleomycin-induced IPF murine models [66].

In 2019, in a more mechanistic approach, we described that the ribonucleoprotein complex MiCEE, formed by the interaction between the nuclear miRNA lethal 7d *MIRLET7D*, the exosome-associated protein C1D, the nuclear-specific exosome subunit EXOSC10, and the histone methyl transferase enhancer of zeste homolog 2 EZH2, mediates the epigenetic silencing of bi-directionally expressed genes in non-fibrotic cells [7]. This mechanism, regulated by the MiCEE complex, is compromised during fibrogenesis due to reduced *MIRLET7D* levels. In consequence, high transcript levels of fibrotic markers (e.g., *FN1*, *PSMD3*, *SNAI1*, *COL1A1*, *NOLC1*, *SNAI2*, *COL3A1*, *MAFG*) in lung tissue samples from IPF patients compared to controls were reported. Remarkably, we proposed that reduced *MIRLET7D* and HDAC1/2 nuclear activity mediated by hyperactive EP300 disrupts the MiCEE complex and interferes with its proper function, promoting fibrosis formation. As a pharmacological confirmation, EP300 inhibitor treatment attenuated fibrosis effects in the bleomycin mouse model (in vivo), in patient-derived primary fibroblast (in vitro), and in precision-cut lung slices (PCLS) from patients (ex vivo), with a consistent reduction in extracellular matrix protein deposition, fibrotic markers (e.g., FN1, COL1A1, and ACTA2), cell migration, and proliferation. These findings, highlighted by complementary experimental approaches, suggest EP300 inhibition as a promising target in IPF [7].

### 4.4. Liver Fibrosis

Liver fibrosis is a wound-healing response that occurs during chronic liver injury and viral hepatitis. It results from collagen and other ECM proteins accumulation between hepatocytes and the sinusoidal endothelium. These alterations in ECM composition promote disruption of normal liver physiology and organ architecture. Collagen-producing cells in the injured liver include activated hepatic stellate cells (HSC), portal fibroblasts, and myofibroblasts of bone marrow origin [67]. HSC are pleiotropic non-parenchymal cells continuously activated in response to cytokines, reactive oxygen species, and other mediators of inflammation by autocrine and paracrine stimulation.

Persistent hepatic fibrosis may progress to cirrhosis, liver failure, portal hypertension, hepatocellular cancer, and, unfortunately, death [68,69]. In 2021, Wang et al. obtained a transcription profile by single-cell RNAseq (sc-RNAseq) in fibrotic and cirrhotic compared to healthy human livers, using an in vivo model of thioacetamide (TAA)- and bile duct ligation (BDL)-induced liver fibrosis in rat models [70]. Their results showed that differentially expressed genes after activation of HSCs corresponded to fibrosis markers such as *COL1A1*, *COL1A2*, *DCN*, and *LUM*. Interestingly, they identified two master regulators of HSC activation, RUNX1 and CREB3L1, this last one including target genes such as *COL1A1*, *LOX*, *MMP14*, *TGFB3*, *TIMP*, and *VCAN*.

Other genes that have been implicated in fibrogenesis include *AEBP1*, *PRRX1*, and *LARP6* in nonalcoholic steatohepatitis (NASH) with fibrosis. Moreover, these authors determined two potential ligands for the epidermal growth factor receptor (EGFR) during HSC activation: amphiregulin (AREG) and progranulin (GRN), which are relevant mechanisms in immune cells such as Kupffer cells and monocyte-derived macrophages since they secrete high amounts of both stimulating molecules for EGFR signaling activation [70].

In addition, dimethyl α-ketoglutarate treatment inhibits in vitro HSC activation via autophagy suppression in the immortalized rat liver stellate cell line HSC-T6 and in vivo in a carbon tetrachloride (CCl4)-induced liver fibrosis model [71]. These experiments suggested that dimethyl α-ketoglutarate increases cytoplasmic levels of α-ketoglutarate, which is transformed to acetyl-CoA. Acetyl-CoA catalyzes EP300 acetylation, and in consequence, some autophagy-related proteins are acetylated by EP300, blocking autophagy and subsequently HSC inactivation [71]. Interestingly, primary murine and human HSCs seeded in high stiffness hydrogel developed α-SMA-positive stress fibers, relating to HSCs, and EP300 nuclear accumulation compared to lower stiffness (0.4 kPa).

In addition, it was proposed that RHOA is required for stiffness-mediated phosphorylation of AKT and EP300 and the concomitant transcription of more than 20 tumor-promoting factors, including *CXCL12*, *IL11*, *IL6*, *VEGFA*, *PDGFA* and *PDGFB*, *FGF*, and *CTGF*. Therefore, these elegant approaches determined that culture stiffness influences EP300’s role as a mechanosensitive molecule mediating mechanotransduction of HSC activation, tumorigenesis, and tumor progression.

## 5. Epithelial-to-Mesenchymal Transition and TGFβ Crosstalk

One fundamental molecular mechanism for the development and progression of fibrosis is the epithelial-to-mesenchymal transition (EMT), in which epithelial cells reprogram their phenotypes toward myofibroblasts. Initially, EMT takes place through the loss of epithelial markers such as E-cadherin and cytokeratin, the nuclear transformation of β-catenin, and the novel expression of mesenchymal markers including vimentin, α-smooth muscle actin, and fibroblast-specific protein 1. This process probably arises in cooperation with hypoxia-inducible factors, integrins, and the ECM itself. TGFβ, through the SMAD pathway and its interplay with regulatory miRNAs, is the master modulator of EMT in fibrotic and other hyperproliferative diseases [46,72,73].

A cohort study including paroxysmal (<7 days) versus persistent (>7 days) atrial fibrillation patients proved that altered expression of *TGFβ1*, *MMP-9*, and tissue inhibitor of metalloproteinase-1 (*TIMP-1*) plays a vital role during fibrosis progression, whereas *IL-6* contributes to inflammation events [74]. In this sense, Goldmann et al. (2018) evaluated the transcriptome of primary human alveolar epithelial cells type II (hAECII) treated with TGFβ or TGFβ-inhibitor [75]. They identified an exclusive TGFβ fingerprint formed by 454 genes up-regulated by TGFβ that interact with actin cytoskeleton remodeling, differentiation processes, and collagen metabolism. Also, a total of 53 genes were up-regulated in a parallel manner in IPF and in TGFβ-treated hAECII by EMT events, myogenesis, collagen formation, and ECM architecture [75].

Additionally, it was demonstrated that higher serum level and activity of tissue nonspecific alkaline phosphatase (TNAP) correlated to a ST-segment elevation in myocardial infarction patients. TNAP and α-SMA were significantly upregulated in TGFβ1-activated primary cultures of neonatal rat cardiac fibroblasts (CFs). It is known that SMAD2 phosphorylation (Ser465/467) is enhanced by TGFβ1. In contrast, TNAP inhibition was associated with the phosphorylation of AMPKα1/2 (Thr183/172) and increasing expression of *Smad7*, a dephosphorylation factor of SMAD2, attenuating TGFβ1 effects. Thus, AMPK activation downregulates TGFβ1/SMAD2 signaling by activating SMAD7. Moreover, TNAP inhibition promotes the upregulation of TP53, which attenuates migration, differentiation, and expression of collagen-related genes in TGFβ1-activated CFs. Taken together, TP53 signaling mediates the antifibrotic effects of TNAP inhibition in a TGFβ1/SMAD-dependent manner in CFs [76].

In terms of epigenetic modulators, the increased expression of DNA methyltransferases 3A and 1 (*DNMT3A* and *DNMT1*) was observed in TGFβ–activated fibroblasts from fibrotic skin patients with systemic sclerosis (SSc), a prototypical idiopathic fibrotic disease. In fact, SMAD inhibited suppressor of cytokine signaling 3 (SOC3)-induced promoter hypermethylation and blocked STAT protein phosphorylation. In this way, downregulation of *SOCS3* is associated with STAT3 activation and fibroblast-to-myofibroblast transition, collagen release, and fibrosis in vitro and in vivo [77]. In concordance with these results, either genetic or pharmacological inactivation of DNMT3A reversed TGFβ-dependent fibroblast activation by STAT3 signaling attenuation in murine models [77].

In addition, a member of class Ⅲ histone deacetylase, SIRT1, was shown to be elevated in a bleomycin-induced lung fibrosis murine model, IPF patients, and human primary lung cell lines (MRC-5) treated with TGFβ1 [78]. Moreover, its activation attenuated TGFβ1-mediated lung fibroblast activation through the regulation of canonical TGFβ/EP300 signaling, thus exerting promising anti-fibrotic effects [78]. In another study performed in cardiac fibrosis, it was demonstrated that phospho-SMAD2/3 recruits EP300 to promoter regions of pro-fibrotic factors [79]. Interestingly, AMPK activation through Ca^2+^/calmodulin-dependent protein kinase β (CaMKKβ) phosphorylation reduced the SMAD3/EP300 interaction, therefore suppressing cardiac fibrosis by baicalin treatment, a flavone found in the fungus Baical skullcap (Scutellaria baicalensis Georgi) roots [79].

In 2018, a revelatory approach showed that the EMT inducibility that is dependent on the molecular axis GLO1/TGFβ1/SMAD, by loss of the tumor suppressor *miR-101*, could be reversed by metformin administration in human invasive cells [80]. Interestingly, in recent years, metformin has been reported by several authors to target EP300 by at least different mechanisms including the following: (1) inhibition of H3K9 acetylation activity [81], (2) blocking the formation of E300-inducible phase separation condensates [82], or (3) direct modulation of EP300 mRNA levels [83], which will surely contribute to novel anti-proliferative and anti-invasive therapies based on EP300-dependent molecular mechanisms.

## 6. EP300 Inhibitors

CBP/EP300 inhibitors can be divided into three classes: inhibitors of the HAT activity, inhibitors of the bromodomain, and degraders; as depicted into categories in Table 1.

### 6.1. Pharmacological EP300 HAT Inhibitors

#### 6.1.1. C646

C646 was one of the first small molecules to inhibit CBP/EP300 HAT activity within the nanomolar range. It is widely used in cancer research. It was selective when compared to the other six HATs: AANAT, PCAF, GCN5, Sas, Moz, and Rtt109 [99]. The treatment of human cardiac fibroblasts with C646 blocks TGFβ-induced collagen synthesis [86]. In SIRT3-KO mice with coronary microvascular dysfunction models, C646 alleviated cardiac remodeling, including cardiac fibrosis, hypertrophy, and capillary rarefaction [89]. On streptozotocin-induced diabetic C57BL/6J mice, C646 down-regulated diabetes-induced pro-fibrotic molecules such as collagen IV, fibronectin, and laminin and significantly reduced diabetes-induced glomerular hypertrophy derived from the accumulation of the ECM proteins [90]. However, C646 has also been reported to display non-specific thiol reactivity, binding to tubulin and inhibiting its polymerization [100,101].

#### 6.1.2. A485

A485 is an orally bioavailable drug with CBP/EP300 HAT inhibitor capacity that is highly potent and selective and has been demonstrated to actively inhibit CBP/EP300 catalytic activity in vitro and in vivo, making it an excellent drug candidate for clinical trials. A485 inhibited tumor growth in a prostate cancer castration-resistant xenograft model [90]. In myofibroblasts from patients with Dupuytren’s disease (DD), a localized fibrotic disorder of the palm, A485 inhibited the profibrotic genes *ACTA2* and *COL1A1* expression [91].

#### 6.1.3. (E)-3-(3-(4-((3-Carbamoylbenzyl) oxy)-3-iodo-5-methoxyphenyl)acryloyl)benzamide(A6)

A6 is a potent CBP/EP300 HAT inhibitor, and it is a suitable drug candidate with better pharmacokinetic properties than C646. A6 was found to have effective anti-fibrotic activity both in vitro on TGFβ1-stimulated lung fibroblasts and in vivo on bleomycin-induced lung fibrosis in mice. A6 reduced the gene expression levels of profibrotic markers and epithelial-to-mesenchymal transition (EMT)-inducible transcription factors (*SNAI1*, *SNAI2*) and decreased the endogenous and trichostatin A-induced histone acetylation levels. Also, it efficiently reduced COL1A1, COL1A2, and FN deposition in lung parenchyma [86]. Moreover, the role of A6 was investigated in liver fibrosis using two mouse models: male mice on a choline-deficient, high-fat diet (CD-HFD) and thioacetamide (TAA). It was found that A6 treatment significantly reduced tissue pathological hallmarks of liver fibrosis as well as ameliorated the TGFβ1-induced increase in fibrotic proteins COL1A, CTGF, FN, TNC, and periostin. They also demonstrated that A6 treatment improved liver fibrosis by reducing the stability of CBP/EP300 via disruption of CBP/EP300 binding to AKT [87].

#### 6.1.4. L002

L002 has been probed in an angiotensin II (Ang II)-induced cardiac fibrosis mouse model as a potential therapeutic option for hypertensive cardio-renal fibrosis [84]. Mice administered with L002 exhibit reductions in cardiac fibrosis, hypertrophy, and renal fibrosis. In vitro, the inhibition of HAT activity by L002 reduces TGFβ-mediated cellular migration, proliferation, ECM protein synthesis (collagen, α-SMA), and CBP/EP300 upregulation [85]. Overall, the effects of CBP/EP300 inhibition by L002 appear to be context-dependent and can vary depending on the specific cellular and molecular mechanisms involved. Further research is needed to fully understand the potential therapeutic applications of L002, considering its lack of specificity when inhibiting EP300/CBP-associated factor (PCAF) and GCN5, two proteins of the GNAT (GCN5-related N-acetyltransferase) family [84].

### 6.2. Natural Products as EP300 HAT Inhibitors

#### 6.2.1. Plumbagin

Plumbagin is a natural compound found in the roots of the medicinal plant *Plumbago zeylanica* and has anti-inflammatory and anti-cancer properties. Plumbagin has been reported to be a potent HAT inhibitor of CBP/EP300 but is not active against GCN5 or PCAF, indicating that plumbagin is a selective inhibitor against CBP/EP300 [94]. Plumbagin suppressed TGFβ-induced profibrotic gene expression and proliferation in fibroblasts. Moreover, plumbagin significantly inhibited bleomycin-induced pulmonary fibrosis in mice. Overall, inhibition of the CBP/EP300 histone acetyltransferase activity by plumbagin appears to be a potential therapeutic approach for managing pulmonary fibrosis [95]. However, there are several challenges to the use of plumbagin as a therapeutic agent, for instance, the lack of any approved medical trial for plumbagin in humans. In addition, this drug’s limited solubility and oral bioavailability, together with some reports of toxicity, highlight the necessity of further research in this field [102].

#### 6.2.2. Curcumin

Curcumin is a natural compound found in turmeric. Curcumin has been shown to inhibit the development of left ventricular hypertrophy (LVH) by suppressing CBP/EP300-HAT activity and, in consequence, attenuating the acetylation levels of *GATA4* in the cardiac tissue of Dahl salt-sensitive hypertensive rats [93]. This report also showed a significant reduction in hypertension-induced increases in myocardial cell diameter, perivascular fibrosis, and hypertrophy-response gene expression, including *Anf* and *Myh7* (previously *β-MHC*) [93]. Also, in the case of atrial fibroblast senescence and age-related atrial fibrosis, some research has shown that curcumin can inhibit the activity of CBP/EP300-HAT and the TP53/SMAD3 axis. This may lead to a reduction in the accumulation of senescent cells and the development of fibrosis in the atria. Additionally, curcumin has been shown to have direct anti-fibrotic effects in the cardiac fibrosis of aged animal models, which showed improved electrophysiological characteristics and reduced atrial fibrillation susceptibility [59]. More research is needed to determine whether these effects may translate to humans since there are currently no approved clinical applications or clinical trials for curcumin as a CBP/EP300 inhibitor. However, curcumin is widely available as a dietary supplement and is commonly used as a natural anti-inflammatory and antioxidant agent. Such considerations should be made for natural products in general, mostly because they exert pleiotropic effects on cells that must be systematically evaluated.

### 6.3. Pharmacological EP300 Bromodomain Inhibitors

Bromodomains can be divided into two main classes: (1) bromodomain and extra-terminal (BET) and (2) non-BET. Extensive research using small-molecule bromodomain inhibitors has historically focused on the BET class and only recently on non-BET domain inhibitors. Special emphasis has been placed on the drug selectivity of >30-fold over the BET bromodomain class (Table 1).

#### 6.3.1. SGC-CBP30 and I-CBP112

SGC-CBP30 and I-CBP112 were two of the first selective CBP/EP300 bromodomain inhibitors within the nanomolar range [96,97]. In myofibroblasts from Dupuytren’s disease patients, SGC-CBP30 and I-CBP112 were used to validate CREBBP and EP300 as key regulators of the profibrotic phenotype in myofibroblasts by decreasing the expression of the profibrotic genes *ACTA2* and *COL1A1* [92]. Specifically, SGC-CBP30 identified collagen VI as a prominent downstream regulator of myofibroblast activity [91]. On an in vitro large-scale screen of compounds, I-CBP112 and SGC-CBP30 demonstrated their ability to block the fibrogenic differentiation of muscle-resident fibro/adipogenic progenitors [103]. Moreover, they found in both compounds almost a 50% tendency to block collagen levels induced by TGFβ1 in vitro [103].

CCL2-dependent infiltrating monocytes and macrophages promote liver fibrosis by increasing angiogenesis and local immune cell-mediated cytotoxic injury [2]. SGC-CPB30 can abrogate CBP/EP300-dependent *CCL2* transcription in liver sinusoidal endothelial cells (LSECs) induced by TNFα, and associated macrophage chemotaxis in vitro. The authors concluded that SGC-CBP30 prevented the recruitment of the EP300/BRD4/NFκB Icomplex to the *CCL2* enhancer and promoter regions [2]. Furthermore, SGC-CBP30 reduces fibrotic hallmarks in vitro (patient-derived primary fibroblast), in vivo (bleomycin mouse model), and ex vivo (precision-cut lung slices, PCLS) in idiopathic pulmonary fibrosis models, in particular by reducing extracellular matrix protein deposition, levels of fibrotic markers such as FN1, COL1A1, and ACTA2, cell migration, and proliferation [7].

SGC-CBP30 also proved to be efficient in inhibiting IL-17A secretion by Th17 cells from healthy donors and patients with ankylosing spondylitis and psoriatic arthritis [104], results that are relevant due to the involvement of this cytokine in the pathogenesis of lung fibrosis [105]. In contrast, a report suggested that I-CBP112 was likely to activate CBP/EP300 in an allosteric manner through bromodomain interactions, enhancing H3K18 acetylation in acute leukemia and prostate cancer cells within a concentration range effective for its antiproliferative effects [106].

Interestingly, a mechanism of action achieved by SGC-CBP30 might involve disruption of EP300–lncRNA interactions. As reported by Navarro-Corcuera et al. [107], EP300 forms a functional complex with ELK1 to interact with *ACTA2-AS1* in a lipopolysaccharide-inducible liver fibrosis model. Abrogation of EP300 activity was as efficient to attenuate fibrosis in vitro and in vivo as the cholangiocyte-selective *EP300* KO, in a *ACTA2-AS1* dependent manner, which indicates the reversible cooperation between EP300 and lncRNAs during liver fibrosis [107].

#### 6.3.2. Clinical Trials with CBP/EP300 Bromodomain Inhibitors

The recent discovery of potent and selective CBP/EP300 inhibitors has been key to decoding CBP/EP300’s role in biology and could soon be paramount as drug candidates to develop new fibrosis treatment strategies. In the cancer field, FT-7051, which is an oral, potent, and selective inhibitor of CBP/EP300 bromodomain with activity in preclinical prostate cancer models, is currently in a phase 1 open-label study to examine the safety, PK/PD, and preliminary anti-tumor activity in castration-resistant prostate cancer patients (NCT04575766) [108]. Another orally available CBP/EP300 bromodomain inhibitor, CCS1477, has been submitted by CELLCENTRIC to a phase I/IIa study to evaluate the safety and efficacy of CCS1477 as monotherapy and in combination in patients with advanced hematological malignancies and solid/metastatic tumors (NCT04068597, NCT03568656) [109]. More CBP/EP300 bromodomain inhibitors are currently under development using structure-based design and optimization of promising lead compounds [110].

Combination therapy of HAT-CBP/EP300 and bromodomain-CBP/EP300 inhibitors is an area of active research in the field of cancer therapeutics and holds promise for improving outcomes for patients. On prostate cancer cells, the combination of I-CBP112 and A-485 was able to synergize to inhibit cancer cell proliferation. The combination treatment led to a significant reduction in CBP/EP300 chromatin occupancy and a corresponding reduction in the expression of specific mRNAs, including androgen-dependent and pro-oncogenic prostate genes such as *KLK3* and *c-MYC*. So far, preliminary results suggest that the pharmacologic advantage of targeting multiple domains within a single epigenetic modification enzyme can lead to improved therapeutic outcomes [106].

### 6.4. EP300 Degraders

To completely ablate the CBP/EP300 function, CBP/EP300 degraders have been designed as heterobifunctional molecules termed proteolysis-targeting chimeras (PROTAC). A chimera consists of a specific target protein ligand linked to the ligand of an E3 ligase such as cereblon (CRBN). The PROTAC bridges the target protein to an E3 ubiquitin ligase, which polyubiquitinates the target protein and directs it to the proteasome for degradation and recycling. Currently, there are no reports of CBP/EP300 PROTACs use in fibrosis models; however, results in different cancers are promising. The PROTAC dCBP-1 is based on the CBP/EP300 ligand GNE-781, a recently described CBP/EP300 bromodomain inhibitor that is potent and selective against multiple myeloma cells by abolishing the *MYC* enhancer [111]. The PROTAC JQAD1 is based on A485, which preferentially depletes full-length EP300 in vitro and in vivo with limited toxicity to normal tissues, where CBP may compensate. EP300 degradation results in the loss of H3K27ac at chromatin, loss of the dominant neuroblastoma oncoprotein MYCN, suppression of core regulatory circuitry-based transcription, and apoptosis [112].

## 7. Conclusions and Perspectives

The role of EP300-mediated epigenetic regulation in the pathogenesis of fibrosis has been extensively investigated in recent years, with numerous studies highlighting the potential therapeutic value of targeting EP300 and its associated signaling pathways. While much remains to be learned about the precise mechanisms by which EP300 mediates epigenetic regulation in fibrosis, recent advances in the field have paved the way for a promising future in which the development of novel therapeutics targeting EP300 may provide effective treatments for fibrosis.

One promising avenue for future research is the identification of specific molecules that can modulate the activity of EP300 with no severe side effects. Several studies have shown that small molecules, such as C646 and A-485, can selectively inhibit the acetyltransferase activity of EP300, leading to the attenuation of fibrosis-related processes. However, the development of safe and effective EP300 inhibitors for clinical use remains a challenge. Future studies may focus on the development of novel EP300 inhibitors with increased specificity and improved pharmacological properties.

Another promising approach is the use of gene editing technologies, such as CRISPR/Cas9, to target *EP300* and other genes involved in fibrosis. Recent studies have demonstrated the feasibility of using CRISPR/Cas9 to target *EP300* and other genes involved in fibrosis-related processes, resulting in the attenuation of fibrosis in animal models. Moreover, the development of CRISPR-based therapies for human diseases is rapidly progressing, with several clinical trials underway [113,114,115]. In addition to targeting EP300 directly, targeting its associated signaling pathways may also offer potential therapeutic value. For example, the TGFβ signaling pathway plays a crucial role in the development and progression of fibrosis and has been shown to be regulated by EP300. Targeting the TGFβ signaling pathway using small molecule inhibitors or biologics may provide effective treatments for fibrosis. Furthermore, the use of EP300-mediated epigenetic regulation as a biomarker for the diagnosis and prognosis of fibrosis could be an essential aspect of future research. Several studies have shown that the expression of EP300 is dysregulated in fibrosis, and targeting EP300 can attenuate fibrosis-related processes. Therefore, measuring the expression levels of EP300 or its associated epigenetic modifications may provide valuable information for the diagnosis and prognosis of fibrosis. However, further research is needed to determine the validity and reliability of EP300-mediated epigenetic regulation as a biomarker for fibrosis.

Finally, the development of personalized medicine approaches for the treatment of fibrosis is another potential future perspective. Fibrosis is a complex disease with multiple etiologies, and different patients may respond differently to the same therapy. Therefore, developing personalized medicine approaches that consider the individual patient’s genetic and epigenetic profile may lead to more effective and personalized therapies for fibrosis. EP300-mediated epigenetic regulation may represent a potential target for personalized medicine approaches, as the expression of EP300 and its associated epigenetic modifications may vary between patients with fibrosis. In conclusion, the identification of novel therapeutic targets for the treatment of fibrosis represents a promising area of research, and targeting EP300-mediated epigenetic regulation is particularly attractive toward developing a successful strategies. Further research is needed to fully elucidate the mechanisms by which EP300 mediates epigenetic regulation in fibrosis and to develop safe and effective therapeutics targeting EP300 and its associated signaling pathways. Nonetheless, the growing understanding of the role of EP300 in fibrosis and the advances in the development of novel therapeutics suggest a bright future for the treatment of this devastating disease.

## Figures and Tables

**Figure 1 ijms-24-12302-f001:**
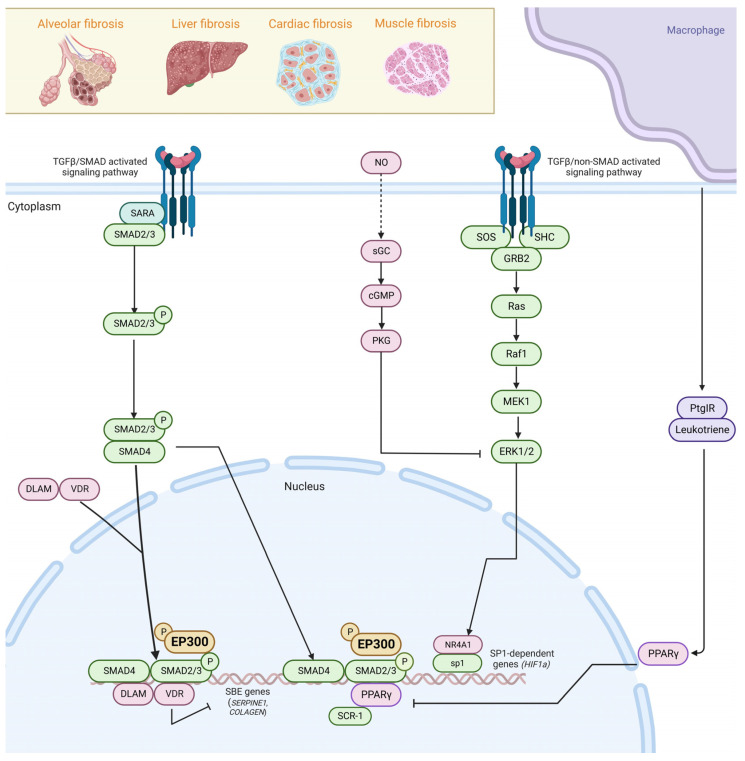
EP300 mediates epigenetic regulation in multi-organ fibrosis by canonical and non-canonical TGFβ pathway activation. In canonical TGFβ signaling, TGFβ ligand binds to TGFβRI (TβRI) and TGFβRII (TβRII) receptors, resulting in their activation in a hetero-tetrameric complex. Activation via the TGFβRI receptor can lead to systemic fibrotic disease in organs such as the lung, liver, heart, and muscle through canonical TGFβ–SMAD, ERK, JNK, and p38 MAPK, and Rho/ROCK/NF-kappa B signaling pathways. SMAD2/3 are phosphorylated upon TGFβRI activation to form a complex with SMAD4 and with certain transcription factors (TF) to activate pro-fibrotic genes with SMAD-binding elements (SBEs). NR4A1, VDR, and PPARγ inhibit TGFβ signaling by repressing SP1-dependent profibrotic genes and competing with SMAD4 for the binding of SMAD2/3 dimers, limiting TGFβ fibrotic transcriptional programs. TGFβ signaling also induces EMT by activating non-SMAD pathways such as Rho GTPase, PAR6, SOS, and GRB2. Created with biorender.com.

**Figure 2 ijms-24-12302-f002:**
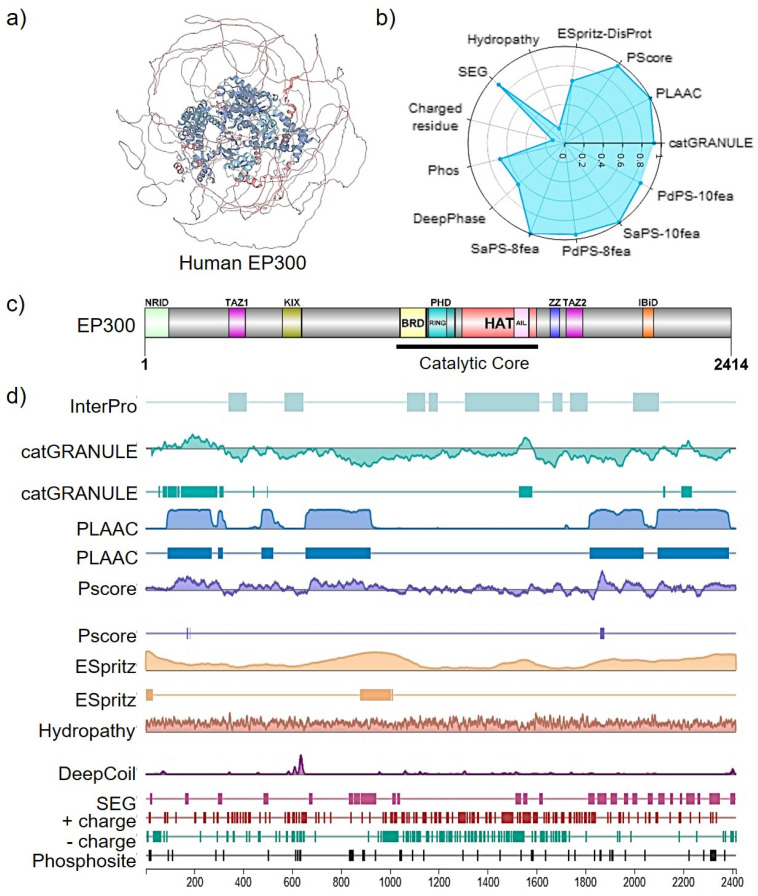
EP300 structural insights (Uniprot Entry Q09472). (**a**) MobiDB manual annotation of intrinsic disorder domains (IDR, colored as alpha helices in blue) in EP300 3D folding derived from experimental data such as X-ray and NMR chemical shifts. (**b**) A radar chart displaying the ranking of disordered features for EP300 evaluated in the human proteome; ESpritz-DisProt: 0.648; Pscore: 0.968; PLAAC: 0.999; catGRANULE: 0.925; PsPS-10fea: 0.888; SaPS: 0.991; PdPS-8fea: 0.952; SaPS-8fea: 0.999; DeepPhase: 0.643; Phos: 0.688; Charged residue: 0.128; SEG: 0.911; Hydropathy: 0.162. A 1-ranking was shown for each feature value (the highest rank score is 1, and the lowest is 0). (**c**) Domain structure of human EP300. (**d**) PhaSePred protein feature viewer and LLPS-related predictions with residue-level scores. PLAAC (Prion-Like Amino Acid Composition); PS-Self score (proteins that can self-assemble to form condensates); PS-Part score (proteins whose phase separation behaviors are regulated by protein or nucleic acid partner components); SaPS (self-assembling phase-separating predictor); PdPS (partner-dependent phase-separating predictor); catGRANULE (granule-forming propensity predictor); PLD (prion-like domain); PScore (π-contact predictor); ESpritz (IDR predictor); SEG (low-complexity region, LCR); CIDER (hydropathy prediction); DeepCoil (coiled-coil domain predictor DeepCoil). Adapted from mobidb.org.

**Table 1 ijms-24-12302-t001:** EP300 inhibitors in fibrotic diseases.

Compound	IC50/Kd	Selectivity	Model	Molecular Mechanism	Fibrotic Phenotype	Ref.
Inhibitors of HAT Activity						
L002	1.98 µm ^a^	3.7-fold more selective for PCALF and GCN5 than for EP300	Ang II-induced hypertense mice		Reduction in cardiac fibrosis, hypertrophy, and renal fibrosis; Reverses hypertension-induced myofibroblast differentiation in murine ventricles.	[5,84,85]
			Human and rat cardiac fibroblasts, human podocytes, human mesangial cells + TGFβ	Inhibits profibrogenic AT1 receptor and partially rescues suppression of antifibrogenic AT2 receptor in human cardiac fibroblasts; Suppresses ERK1/2 MAPK pathway in human mesangial cells; Suppresses phosphorylation of pSMAD2 and pERK1/2 (canonical and non-canonical TGFβ signaling) in rat cardiac fibroblasts.	Reduces TGFβ-mediated cellular migration, proliferation, ECM protein synthesis (collagen, α-SMA), and CBP/EP300 upregulation. Blocks TGFβ-induced fibroblast-to-myofibroblast differentiation (reduction in αSMA); Suppresses pro-fibrogenic responses in cardiac fibroblasts, podocytes, and mesangial cells.	
A6	0.87 µM ^a^		Bleomycin-induced mouse model of lung fibrosis	Reduced the level of activated TGFβ1 in the bronchoalveolar lavage fluid.	Reduced collagen deposition in the lung parenchyma.	[86,87]
			Liver fibrosis mouse models (choline-deficient/high-fat diet and thioacetamide models)		Reduced liver fibrosis; Reduced fibrosis marker genes (smooth muscle actin (*αSMA*), tenascin C, collagen type 1, collagen type 3, connective tissue growth factor).	
			Hepatic stellate cells + TGFβ	Dissociation of EP300 from AKT.	Decreasing levels of TGFβ1-induced αSMA and fibrosis markers COL1A, CTGF, fibronectin (FN), TNC, and periostin.	
			Mouse lung fibroblast + TGFβ1	Epithelial-to-mesenchymal transition (EMT)-inducible transcription factors (SNAI1, SNAI2) and decrease in the endogenous and trichostatin A-induced histone acetylation level.	Reduced expression of *COL1A1*, *COL1A2*, *FN* and *ACTA2*.	
C646	1.6 µM ^b^		Human cardiac fibroblasts + TGFβ		Blocks collagen synthesis.	[85,88,89]
			Streptozotocin-induced diabetic mice		Reduced glomerular hypertrophy by down-regulating diabetes-induced pro-fibrotic molecules (collagen IV, fibronectin, and laminin).	
			Coronary microvascular dysfunction mice model (SIRT3KO)	Suppressed TLR-4/IRAK-4/MyD88–mediated NF-κB	Attenuated cardiac remodeling (cardiac fibrosis, hypertrophy).	
A485	9.8 nM ^c^		Myofibroblasts from patients Dupuytren’s disease		Inhibited profibrotic genes *ACTA2* and *COL1A1* expression.	[90,91]
Curcumin	∼25 μM ^b^		Dahl salt-sensitive hypertensive rats	Attenuated acetylation levels of GATA4	Inhibited left ventricular hypertrophy development; Reduced increases in myocardial cell diameter, perivascular fibrosis and transcription of the hypertrophy-response genes, including *Anf* and *Myh7*.	[59,92,93]
			Aged mice	Inhibited TP53/SMAD3 axis activity	Rescued senescence and fibrosis in the atrial tissue, and atrial fibrillation vulnerability.	
			Senescent human atrial fibroblasts	Inhibited TP53/SMAD3 axis activity	Fibrosis proteins COL1A1, MMP2/9, and TGFβ levels were reduced.	
Plumbagin	20 µM ^b^		Fibroblast + TGFβ		Inhibited fibroblast proliferation and fibrotic targets transcription: *COL1A1*, *COL3A1*, *ACTA2 FN*, *SNAIL*, and *SERPINE1*.	[94,95]
			Bleomycin-induced mouse model of lung fibrosis		Inhibited pulmonary fibrosis	
**Inhibitors of bromodomain (Non-BET)**						
SGC-CBP30	32 nM ^d^	40-fold ^e^	Myofibroblasts from patients Dupuytren’s disease	Specifically, SGC-CBP30 identified collagen VI (Col VI) as a prominent downstream regulator of myofibroblast activity.	Inhibited profibrotic genes *ACTA2* and *COL1A1* expression.	[2,7,91,96]
			Liver sinusoidal endothelial cells (LSECs) + TNFα	Prevented recruitment of EP300/BRD4/NFκB complex to the CCL2 enhancer and promoter regions.	Abrogated CBP/EP300-dependent CCL2 transcription and associated macrophage chemotaxis in vitro.	
			Idiopathic pulmonary fibrosis models: patient-derived primary fibroblast, bleomycin mouse model, and ex vivo (precision-cut lung slices).	Reconstituted HDAC activity and MiCEE function in IPF fibroblasts	Reduced fibrotic hallmarks: matrix protein deposition, levels of fibrotic markers, such as FN1, COL1A1, and ACTA2, and reduced cell migration and proliferation.	
I-CBP112	170 nM for CPB ^f^	37 fold ^e^	Myofibroblasts from patients Dupuytren’s disease.		Inhibited profibrotic genes *ACTA2* and *COL1A1* expression.	[91,97,98]

^a.^ IC50 Fluorescence-based p300 acetyltransferase activity assay. ^b.^ IC50 14C-acetyl-CoA transfer to Ac-H4-15 assay. ^c.^ IC50 time-resolved fluorescence resonance energy transfer. ^d.^ Kd isothermal titration calorimetry. ^e.^ Fold selective for CBP/p300 over BRD4 (1). ^f.^ IC50 Alpha screen.

## Data Availability

Not applicable.

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
