# Peer review of "EP300 as a Molecular Integrator of Fibrotic Transcriptional Programs"

_ijms, 2023, doi:10.3390/ijms241512302_

Round 1

Reviewer 1 Report

EP300 as a molecular integrator of fibrotic transcriptional programs

he main topic of the article is related to the fibrosis process centered on histone acetyltransferase EP300 as an essential regulator of epigenetic changes.
I believe that the subject is relevant in the field. There are other publications in this sense, but not so precisely elaborated (Gong Y, Dou Y, Wang L, Wang X, Zhao Z. EP300 promotes renal tubular epithelial cell fibrosis by increasing HIF2α expression in diabetic nephropathy. Cell Signal. 2022). This article focuses on EP300 as a multi-organ epigenetic regulator of fibrosis. The bibliographic study and the diagrams/tables in the text make a meticulous synthesis of the previous studies in order to develop the personalized treatment of fibrosis. References could be up-dated. The conclusions are consistent with the evidence presented previously.

Author Response

We acknowledge the positive comments from Reviewer 1, as well as all constructive observations that helped improving the quality of our work. We have revised our manuscript including 11 new references as suggested by Reviewers 1 and 3, which were included in Sections 3 (lines 233-241), 4.1 (lines 288-292), 4.2 (320-322) and 5 (lines 496-503). Newly included references have been either recently published and not considered for the first draft of this work, or we have not suficiently discussed specific points along the text which we expect to cover in a more robust way in this version.

Reviewer 2 Report

This review article is covering EP300 a histone acetyltransferase role as a regulator of the epigenetic changes contributed to fibrosis. The literature compilation data presented in the manuscript were, directed for the development of the classification of EP300 inhibitors and dividing them into three classes: inhibitors of the HAT activity, inhibitors of the bromodomain (BET and non-BET) and degraders. 

Additionally, review is decorated with two essential figures and one table and is concluded with 105 very recent literature references. Particularly informative is figure 1, clearly characterizing important functionality of the EP300 epigenetic regulation in multi-organ fibrosis by TGF-b-pathways activation. The authors should be complimented for such an excellent summary of genesis of organ fibrosis diseases.

This will constitute crucially important goals and novelty of this very important paper.

The following suggested changes and recommendations should be introduced before the publication of the manuscript.

1.     Page 2, Line 69. Replace “ importantly” with “effectively”.

2.     Page 4, Line 121. Correct “5-hydroxytyyrosine” to “5-hydroxytyrosine”.

3.     Page 4, Line 149 Insert “totally” in front of “distinct”. 

4.     Page 7, Line 260. Replace “line” with “fashion”.

5.     Page 8, Line 302. Replace “aged” with “mature”.

6.     Page 11, Line 439-443. The sentence “demonstrated that higher” is too long and must be split into two parts.

7.     Page 17, Line 608. Insert missing reference number after Navarro [97].

8.     Page 19, Line 692. Insert “for successful” after “attractive” 

The manuscript is of good quality and urgent importance and is well written and edited in order to meet the standard for the articles published inInternational Journal of Molecular Sciences. Thus, I certainly recommend it for publication after the correction of these suggested minor changes. 

Author Response

We acknowledge the positive comments from Reviewer 2, as well as all constructive observations that helped improving the quality of our work. We have fully addressed all 8 points indicated by Reviewer 2. Moreover, we have edited once more the english grammar along all sections.

In addition, we have revised our manuscript including 11 new references as suggested by Reviewers 1 and 3, which were included in Sections 3 (lines 233-241), 4.1 (lines 288-292), 4.2 (320-322) and 5 (lines 496-503). Newly included references have been either recently published and not considered for the first draft of this work, or we have not suficiently discussed specific points along the text which we expect to cover in a more robust way in this version.

Reviewer 3 Report

This is a comprehnsive and very well organized review about the role of EP300-mediated epigenetic regulation in multi-organ fibrosis and its potential as a therapeutic target. Preclinical evidence are also reported on EP300 inhibition in attenuating fibrosis-related molecular processes, including extracellular matrix deposition, inflammation, and epithelial-to- mesenchymal transition. Finally, the contributions of small molecule inhibitors and gene therapy approaches targeting EP300 are described as novel therapies against fibrosis.

The review is original and very interesting. 

I just have a couple of suggestions for authors to implement references, if they agree, about recent papers they might not know, that could improve the knowledge of mechanisms involved in fibrosis:

Paragraph 2, rows 101-102: maybe, these papers could be of interest for authors here: PMID: 31174324, PMID: 29504694

Paragraph 4.2 rows 337-338: maybe, these papers could be of interest for authors here: PMID: 32756399, PMID: 26784015.

Author Response

We acknowledge the positive comments from Reviewer 3, as well as all constructive observations that helped improving the quality of our work. We have revised our manuscript including 11 new references as suggested by Reviewers 1 and 3, which were included in Sections 3 (lines 233-241), 4.1 (lines 288-292), 4.2 (320-322) and 5 (lines 496-503). Newly included references have been either recently published and not considered for the first draft of this work, or we have not suficiently discussed specific points along the text which we expect to cover in a more robust way in this version.
